# Multimodal AutoML on Structured Tables with Text Fields

**Xingjian Shi**[*]                                    xjshi@amazon.com
**Jonas Mueller**[*]                              jonasmue@amazon.com
**Nick Erickson**                                  neerick@amazon.com
**Mu Li**                                                mli@amazon.com
**Alexander J. Smola**                          alex@smola.org

*Amazon Web Services, CA, USA*

## Abstract

We design automated supervised learning systems for data tables that not only contain numeric/categorical columns, but text fields as well. Here we assemble 15 multimodal data tables that each contain some text fields and stem from a real business application. Over this benchmark[1], we evaluate numerous multimodal AutoML strategies, including a standard two-stage approach where NLP is used to featurize the text such that AutoML for tabular data can then be applied. We propose various practically superior strategies based on multimodal adaptations of Transformer networks and stack ensembling of these networks with classical tabular models. Beyond performing the best in our benchmark, our proposed (fully automated) methodology[2] manages to rank 1st place (against human data scientists) when fit to the raw tabular/text data in two MachineHack prediction competitions and 2nd place (out of 2380 teams) in Kaggle's Mercari Price Suggestion Challenge.

## 1. Introduction

Automatic Machine Learning (AutoML) aims produce end-to-end pipelines that can ingest raw (messy) data, train models, and output accurate predictions, all without human intervention (Hutter et al., 2018). Given their immense potential, many AutoML systems exist for data structured in tables, which are ubiquitous across science/industry (He et al., 2019; Truong et al., 2019; Gijsbers et al., 2019). Many data tables contain not only numeric and categorical fields (together referred to as *tabular* here), but also fields with free-form text. For example, Table 1 depicts actual data from the website Kickstarter. Despite their commercial value, there exist few automated ML solutions for such multimodal data.

In this paper, we consider design choices for automated supervised learning with multimodal datasets that jointly contain text, numeric, and categorical features. Even though text is extremely common in enterprise data tables, how to automatically analyze such multimodal data has not been well studied. This likely stems from a lack of available benchmarks, as well as existing beliefs that basic featurization of the text (Eisenstein, 2018; H2O.ai) should suffice for tabular models to exhibit strong performance. By introducing a new benchmark of 15 multimodal text/tabular datasets from real business applications, we provide the first comprehensive evaluation of different strategies for supervised learning with data of this form.

---

[*]. Equal contribution.

1. Available at: `https://github.com/sxjscience/automl_multimodal_benchmark`

2. Available at: `https://github.com/awslabs/autogluon`

| name | desc | goal | country | currency | created_at | final_status |
|------|------|------|---------|----------|------------|--------------|
| The Secret Order - The Game that gives back Gl... | Can you trust your friends? Solve the puzzle? ... | 5000.0 | GB | GBP | 1424101105 | 0 |
| Booker Family Foods. Home made, the way food s... | Community based, home-made-foods producer, to ... | 2500.0 | US | USD | 1404617242 | 0 |
| J.A.E.S.A : Next Generation Artificial Intelli... | A true next generation AI with the ability to ... | 30000.0 | CA | CAD | 1399078600 | 1 |

Table 1: Example of data in our multimodal benchmark with text (*name*, *desc*), numeric (*goal*, *created_at*), and categorical (*country*, *currency*) columns. From these features, we want to predict if a Kickstarter project will be funded or not (*final_status*).

## 2. Methods

**Featurizing Text for Tabular Models**  In this paper, all tabular (numeric/categorical) modeling is simply done via AutoGluon-Tabular, a highly accurate open-source tool for automated supervised learning on tabular data (Erickson et al., 2020; Yoo et al., 2020). AutoGluon achieves strong performance by ensembling a diverse suite of high-quality models for tabular data. While neural networks are popular for text, decision tree ensembles are typically superior for tabular data (Bansal, 2018; Fakoor et al., 2020b; Huang et al., 2020).

To allow tabular models to access information in text fields, the text is typically first mapped to a continuous vector representation which replaces a text column in our data table with multiple numeric columns (one for each vector dimension). One can treat each text column as a document, and each individual text field as a paragraph within the document, such that each text field can be featurized via NLP methods for computing text representations (Eisenstein, 2018; Devlin et al., 2019; Mikolov et al., 2013).

**Transformer Models for Text**  Pretrained Transformers have become a cornerstone of modern NLP, where the model is first pretrained in an unsupervised manner on a massive text corpus before being applied to our (smaller) labeled dataset of interest (Devlin et al., 2019; Raffel et al., 2020). This allows our modeling to leverage information gleaned from the external text that would otherwise not be available in our limited labeled data. The Transformer also effectively aggregates information from various aspects of a training example, using a *self-attention* mechanism to contextualize its intermediate representations based on particularly informative features (Vaswani et al., 2017).

**Neural Architectures for Multimodal Data**  In many multimodal datasets, some of the predictive signals are solely found in text fields, while other predictive information is restricted to tabular feature values (or interactions between text/tabular values). To enjoy the benefits of end-to-end learning without sacrificing accuracy, we present various strategies to adapt Transformer networks to simultaneously operate on inputs from both modalities.

**All-Text**  A simple (yet crude) option is to convert numeric and categorical values to strings and subsequently treat their columns also as text fields (Raffel et al., 2020). Through its byte-pair encoding, a pretrained Transformer can handle most categorical strings and may be able to crudely represent numeric values within a certain range (here we round all numbers to 3 significant digits in their string representation).

**Fuse-Early**  Rather than casting them as strings, we can allow our model to adaptively learn token representations for each numeric and categorical feature via backpropagation (see Figure 1b). We introduce an extra factorized embedding layer (Lan et al., 2019) to

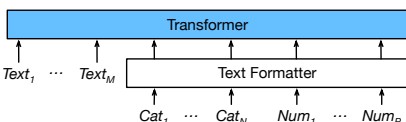

(a) *All-Text*. Convert numeric and categorical values into additional text tokens.

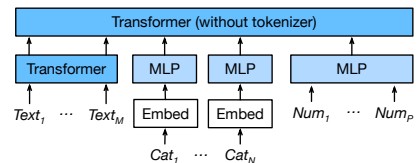

(b) *Fuse-Early*. Transformer operates on learned embeddings for each feature.

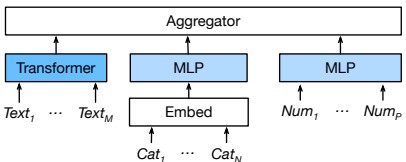

(c) *Fuse-Late*. Separate branches encode each modality, aggregate via mean/max/concat.

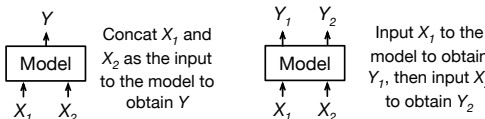

(d) Notation used in these figures.

Figure 1: Fusion strategies in Multimodal-Net, dense output layers on top are not shown.

map categorical values into the same $\mathbb{R}^d$ vector representation encoded by the pretrained NLP model for text tokens (with different embedding layers used for different categorical columns in the table). All numeric features are encoded via a single-hidden-layer Multilayer Perceptron (MLP) to obtain a unified $\mathbb{R}^d$ vector representation. These vectors are fed into a 6-layer Transformer encoder whose self-attention operations can model interactions between the embeddings of text tokens, categorical values, and numeric values.

**Fuse-Late**   Rather than aggregating information across modalities early on in the network, we can perform separate neural operations on each data type and only aggregate near the output layer (see Figure 1c). This design allows each branch to extract higher-level representations of the values from each modality, before the network needs to consider how modalities should be fused. Here we use a multi-tower architecture in which numeric and categorical features are fed into separate MLPs for each modality. The text features are fed into a pretrained Transformer network. Subsequently, the topmost vector representations of all three networks are pooled (via either: mean/max pooling or concatenation) into a single vector from which predictions are output via two additional dense layers.

## 3. Aggregating Text & Tabular Models

Despite their success for modeling text, the application of Transformer architectures to tabular data remains limited (Huang et al., 2020; Fakoor et al., 2020a,b). The use of tabular models together with Transformer-like text architectures has also received little attention (Wan et al., 2021; Ke et al., 2019). Note that 'tabular models' throughout are trained on only numeric/categorical features, e.g., various tree ensembles used in AutoGluon-Tabular.

### 3.1 Embedding Text as Tabular Features

In our first class of aggregation methods, a Transformer is used to map the text fields into a vector representation. Subsequently, the text fields are replaced in the data table by additional columns corresponding to each dimension of the embedding vector (*Embedding-as-Feature* in Figure 2a). We consider three ways to featurize text using a Transformer.

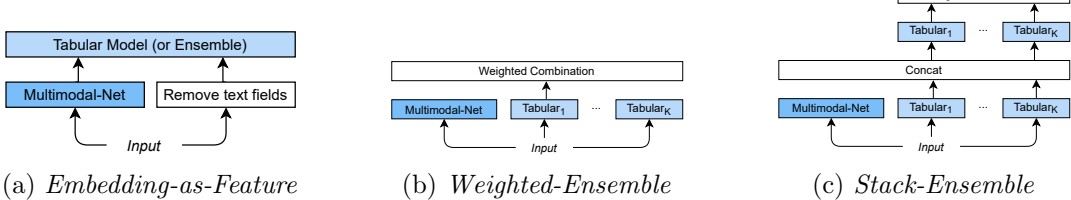

Figure 2: Methods to combine *Multimodal-Net* and classical tabular models.

**Pre-Embedding**   The most straightforward strategy is to use a pretrained Transformer that is not fine-tuned on our labeled data. Subsequently we can train tabular models (or tabular AutoML systems) over the featurized data table (Blohm et al., 2020).

**Text-Embedding**   The Pre-Embedding strategy is not informed about our particular prediction problem and the domain of the text data. In Text-Embedding, we further fine-tune the pretrained Transformer to predict our labels from only the text fields. By adapting to the domain of the problem, Text-Embedding is able to extract more valuable features that can improve the performance of tabular models. This is particularly true in settings where the target only depends on one out of many text fields, since the fine-tuning process can produce representations that vary more based on the relevant field vs. irrelevant text.

**Multimodal-Embedding**   Our text representations may improve when self-attention is informed by context regarding numeric/categorical features. Thus we alternatively consider embedding text via our previous multimodal networks. These models are again fine-tuned using the labeled data and now produce a single vector representation for *all* columns in the dataset, regardless of their type. However, since Transformers are better suited for modeling text than tabular features, we only replace the text fields with the learned vector, all other non-text features are kept and used for subsequent tabular learning.

### 3.2 Ensembling Text/Tabular Predictions

Utilized by most AutoML frameworks (LeDell and Poirier, 2020; Feurer et al., 2015; Erickson et al., 2020), model ensembling is a straightforward technique to boost predictive accuracy. Ensembling is particularly suited for multimodal data, where different models may be trained with different modalities. However, the resulting ensemble may then be unable to exploit nonlinear predictive interactions between features from different modalities. To remedy this, we advocate for the use of our multimodal Transformers that fuse information from text and tabular inputs. Furthermore, we propose stack ensembling with nonlinear aggregation of model predictions that can exploit inter-modality interactions between different base models' predictions, even when base models do not overlap in modality.

**Weighted-Ensemble**   We first consider a straightforward aggregation strategy that simply takes a weighted average of the predictions from our Transformer model and various tabular models like those trained by AutoGluon-Tabular. Here, our Transformer and other models are independently trained using a common training/validation split. Subsequently, we apply *ensemble selection*, a greedy forward-selection strategy to fit aggregation weights over all models' predictions on the held-out validation data (Caruana et al., 2004).

**Stack-Ensemble**   Rather than restricting the aggregation to a linear combination, we can use stacking (Wolpert, 1992). This trains another ML model to learn the best aggregation strategy. The features upon which the 'stacker' model operates are the predictions output by all base models (including our Transformer), concatenated with the original tabular features in the data. Following Erickson et al. (2020), we try each type of tabular model in AutoGluon-Tabular as a stacker model. To output predictions, a weighted ensemble is constructed via ensemble selection applied to the tabular stacker models (Figure 2c). We do not consider our larger Transformer model as a stacker since lightweight aggregation models are preferred in practice. Overfitting is a key peril in stacking, and we ensure that stacker models are only trained over *held out* predictions produced from base models via 5-fold cross-validation (bagging) (Van der Laan et al., 2007; Erickson et al., 2020).

## 4. Experiments

Here we empirically evaluate several multimodal AutoML strategies, with a particular focus on how to best leverage Transformers for text/tabular data. To keep our study tractable, we adopt a sequential decision making process that decomposed our design into three stages: 1) determine the appropriate Transformer backbone and fine-tuning strategy for text data alone, 2) determine the best way for generalizing Transformer to multimodal data among our considered variants, and 3) choose the best method to aggregate text and tabular models. At each subsequent stage of the study, we explore modeling choices that are specific to that stage and simply use the best choice found in the empirical comparisons of the options available in previous stages. Each modeling strategy is run over our benchmark of 15 tabular datasets with text fields, detailed in Appendix E. We evaluate regression tasks via the coefficient of determination ($R^2$), multiclass classification tasks via accuracy, and binary classification tasks via area under the ROC curve (AUC).

**Choice of Transformer Backbone**   Our first decision is which pretrained Transformer network to employ. We consider the base version of RoBERTa (Liu et al., 2019) or ELECTRA (Clark et al., 2020). Existing results may not translate to our setting, since Transformers are typically applied to datasets with at most a couple text fields per training example (Wang et al., 2019b,a). We first fine-tune the pretrained Transformer models as our sole predictors, using only the text features in each dataset. This reveals which model is better for the types of text in our multimodal datasets. In fine-tuning, we consider two tricks to boost performance: 1) Exponentially decay the learning rate of the network parameters based on their depth (Sun et al., 2019); 2) Average the weights of the models loaded from the top-3 training checkpoints with the best validation scores (Vaswani et al., 2017).

   The first section of Table 2 shows that ELECTRA performs better than RoBERTa across the text columns in our benchmark datasets. Our *Text-Net* used in subsequent experiments is thus ELECTRA fine-tuned with both exponential decay and checkpoint-averaging.

**Best Multimodal Network**   Next, we explore the best way to extend the Text-Net model to operate across numeric/categorical inputs in addition to text fields. Three multimodal network variants are considered here: *All-Text*, *Fuse-Early*, *Fuse-Late* (see Figure 1). Across our datasets, Table 2 shows that the *Fuse-Late* strategy outperforms the other options for producing predictions from multimodal inputs using a single neural network (including Text-Net). We fix this *Fuse-Late* model as our *Multimodal-Net* in subsequent experiments.

| Method | prod | qaq | qaa | cloth | airbnb | ae | mercari | jigsaw | imdb | fake | kick | jc | wine | pop | channel | avg. ↑ | mrr ↑ |
|---|---|---|---|---|---|---|---|---|---|---|---|---|---|---|---|---|---|
| Choosing Text-Net: | | | | | | | NLP Backbones and Fine-tuning Tricks | | | | | | | | | | |
| RoBERTa | 0.588 | 0.412 | 0.268 | 0.700 | 0.344 | 0.953 | 0.561 | 0.960 | 0.731 | 0.929 | 0.751 | 0.615 | 0.811 | -0.000 | 0.301 | 0.595 | 0.07 |
| ELECTRA | 0.705 | 0.410 | 0.356 | 0.718 | 0.349 | 0.955 | 0.586 | 0.965 | 0.750 | 0.965 | 0.754 | 0.606 | 0.813 | 0.003 | 0.315 | 0.607 | 0.17 |
| + Exponential Decay $\tau = 0.8$ | 0.728 | 0.436 | 0.431 | 0.743 | 0.337 | 0.953 | 0.579 | 0.963 | 0.852 | 0.963 | 0.760 | 0.664 | 0.808 | 0.004 | 0.308 | 0.635 | 0.09 |
| + Average 3 ★ | 0.729 | 0.451 | 0.432 | 0.746 | 0.350 | 0.954 | 0.581 | 0.965 | 0.858 | 0.961 | 0.766 | 0.656 | 0.807 | 0.004 | 0.307 | 0.638 | 0.12 |
| Choosing Multimodal-Net: | | | | | | | Fusion Strategy | | | | | | | | | | |
| All-Text | 0.907 | 0.454 | 0.419 | 0.746 | 0.366 | 0.957 | 0.599 | **0.967** | 0.840 | 0.967 | **0.799** | 0.645 | 0.810 | 0.013 | 0.480 | 0.665 | 0.19 |
| Fuse-Early | **0.913** | 0.441 | 0.418 | 0.745 | 0.377 | 0.953 | 0.596 | **0.967** | 0.843 | 0.960 | 0.770 | 0.653 | 0.806 | 0.013 | 0.474 | 0.662 | 0.24 |
| Fuse-Late, Concat ★ | 0.907 | 0.449 | **0.445** | 0.747 | 0.395 | 0.958 | 0.603 | 0.966 | 0.857 | 0.961 | 0.773 | 0.639 | 0.812 | 0.015 | 0.481 | 0.667 | 0.17 |
| Fuse-Late, Mean | 0.912 | **0.458** | 0.431 | 0.748 | 0.399 | 0.955 | 0.602 | 0.967 | 0.869 | 0.963 | 0.773 | 0.625 | 0.807 | 0.015 | 0.478 | 0.667 | 0.09 |
| Fuse-Late, Max | 0.910 | 0.452 | 0.429 | 0.747 | 0.401 | 0.956 | 0.599 | 0.966 | 0.863 | 0.957 | 0.761 | 0.634 | 0.808 | 0.015 | 0.484 | 0.665 | 0.12 |
| Choosing Aggregation: | | | | | | | Multimodal Model Ensembling | | | | | | | | | | |
| Pre-Embedding | 0.895 | 0.216 | 0.247 | 0.642 | 0.449 | 0.972 | 0.433 | 0.586 | 0.871 | 0.926 | 0.743 | 0.491 | 0.680 | 0.012 | 0.526 | 0.579 | 0.13 |
| Text-Embedding | 0.867 | 0.446 | 0.432 | 0.748 | 0.430 | 0.972 | 0.434 | 0.587 | 0.855 | 0.962 | 0.790 | 0.658 | 0.830 | 0.008 | 0.502 | 0.635 | 0.20 |
| Multimodal-Embedding | 0.907 | 0.439 | 0.437 | 0.749 | 0.438 | 0.974 | 0.432 | 0.587 | 0.847 | 0.967 | 0.794 | **0.683** | 0.829 | 0.007 | 0.517 | 0.640 | 0.18 |
| Weighted-Ensemble | 0.907 | 0.439 | 0.429 | 0.744 | 0.453 | 0.976 | 0.597 | 0.957 | 0.876 | 0.923 | 0.787 | 0.641 | 0.814 | 0.018 | 0.554 | 0.674 | 0.39 |
| Stack-Ensemble ★ | 0.909 | 0.456 | 0.438 | **0.751** | 0.459 | 0.977 | **0.605** | 0.967 | **0.878** | 0.964 | 0.797 | 0.624 | 0.836 | **0.020** | **0.556** | **0.683** | **0.59** |
| | | | | | | | Tabular AutoML + Feature Engineering Baselines | | | | | | | | | | |
| AG-Weighted | 0.891 | 0.046 | 0.076 | -0.002 | 0.426 | 0.841 | 0.098 | 0.587 | 0.845 | 0.686 | 0.668 | 0.004 | 0.173 | 0.016 | 0.549 | 0.394 | 0.11 |
| AG-Stack | 0.891 | 0.046 | 0.077 | 0.001 | 0.435 | 0.841 | 0.098 | 0.587 | 0.844 | 0.697 | 0.670 | 0.003 | 0.175 | 0.017 | 0.550 | 0.395 | 0.10 |
| AG-Weighted+ N-Gram | 0.892 | 0.426 | 0.382 | 0.610 | 0.450 | 0.978 | 0.526 | 0.909 | 0.842 | 0.966 | 0.772 | 0.357 | 0.829 | 0.019 | 0.546 | 0.633 | 0.11 |
| AG-Stack+ N-Gram | 0.895 | 0.414 | 0.383 | 0.654 | **0.466** | **0.979** | 0.569 | 0.915 | 0.850 | **0.968** | 0.775 | 0.612 | **0.842** | 0.020 | 0.548 | 0.659 | 0.19 |
| H2O AutoML | 0.869 | 0.247 | 0.159 | 0.163 | 0.329 | 0.976 | 0.430 | 0.531 | 0.813 | 0.756 | 0.669 | 0.411 | 0.478 | 0.014 | 0.530 | 0.492 | 0.11 |
| H2O AutoML + Word2Vec | 0.859 | 0.244 | 0.285 | 0.624 | 0.347 | 0.973 | 0.534 | 0.847 | 0.827 | 0.943 | 0.755 | 0.443 | 0.778 | 0.013 | 0.524 | 0.600 | 0.16 |
| H2O AutoML + Pre-Embedding | 0.846 | 0.227 | 0.312 | 0.644 | 0.367 | 0.969 | 0.282 | 0.572 | 0.874 | 0.893 | 0.738 | 0.549 | 0.571 | 0.007 | 0.501 | 0.557 | 0.12 |

Table 2: Predictive performance of AutoML strategies over our multimodal benchmark. Column 'avg.' lists each method's average score (across datasets) and 'mrr' lists the mean reciprocal rank among all models evaluated in the benchmark. Each subsection encapsulates the variants compared at a design stage, with the final choice (best avg.) marked by ★.

**Aggregating Transformers and Tabular Models** Now that we have identified the best single neural network architecture for multimodal text/tabular inputs, we consider how to combine such models with classical learning algorithms for tabular data. Where not specified, the tabular models are those trained by AutoGluon-Tabular (see Appendix C.4). Here we considered the following aggregation strategies: *Pre-Embedding*, *Text-Embedding*, *Multimodal-Embedding*, *Weighted-Ensemble*, *Stack-Ensemble*.

The third section of Table 2 illustrates that *Stack-Ensemble* is overall the best aggregation strategy. As expected, *Text-Embedding* and *Multimodal-Embedding* outperform *Pre-Embedding*, demonstrating how domain-specific fine-tuning improves the quality of learned embeddings. *Multimodal-Embedding* performs better than *Text-Embedding* on some datasets with similar performance across the rest, showing it can be beneficial to use text representations contextualized on numeric/categorical information.

**AutoGluon Baselines** We also compare with variants of AutoGluon-Tabular without our Multimodal-Net as baselines (and variants of H2O AutoML described in Appendix D):

*AG-Weighted / AG-Stack*: We train AutoGluon with weighted / stack ensembling of its tabular models, here ignoring all text columns.

*AG-Weighted + N-Gram / AG-Stack + N-Gram*: Similar to *AG-Weighted / AG-Stack*, except we first use AutoGluon's N-Gram featurization to encode all text in tabular form.

The last section of Table 2 shows that while these powerful AutoML ensemble predictors can outperform our individual neural network models (particularly for datasets with more tabular-signal), our proposed *Stack-Ensemble* and *Weighted-Ensemble* are superior overall[3]. Given the success of pretrained Transformers across NLP, we are surprised to find both N-Grams and word2vec here provide superior text featurization than *Pre-Embedding*.

---

3. Tutorial to easily run these methods on any text/tabular dataset: `https://auto.gluon.ai/stable/tutorials/tabular_prediction/tabular-multimodal-text-others.html`

# Appendix

## Appendix A. Performance in Real-world ML Competitions

Some datasets in our multimodal benchmark originally stem from ML competitions. For these (and other recent competitions with text/tabular data), we fit our automated solution using the official competition dataset, without manual adjustment or data preprocessing. We then submit its resulting predictions on the competition test data to be scored, which enables us to see how they fare against the manual efforts of human data science teams.

Our *Stack-Ensemble* model achieved 1st place in two prediction competitions from MachineHack: *Product Sentiment Classification*[4] and *Predict the Data Scientists Salary in India*[5], and this model achieves 2nd place in another MachineHack competition: *Predict the Price of Books*[6], as well as a Kaggle competition: *California House Prices*[7]. Simply training only our *Multimodal-Net* suffices to achieve 2nd place in a very popular Kaggle competition in which 2380 teams participated: *Mercari Price Suggestion Challenge*[8] (which offered a $100,000 prize). These results demonstrate that, without any manual adjustment, our AutoML solution outperforms top data scientists on real-world text/tabular datasets that possess great commercial value.

## Appendix B. Feature Importance Analysis

Feature importance scoring is a valuable tool for understanding how the AutoML system works and whether text fields in a dataset are worth their overhead. We compute permutation feature importance (Breiman, 2001) for three models: the AG-Stack+N-Gram baseline, our Multimodal-Net, and our Stack-Ensemble (containing the Multimodal-Net). The importance of a feature is defined as the drop in prediction accuracy after values of only this feature (which are entire text fields for a text column) are shuffled in the test data (across rows). We only shuffle original column values so our importance scores are not biased by preprocessing/featurization decisions (except in how these directly affect model accuracy). Figure 3 shows that both Multimodal-Net and our Stack-Ensemble with this model rely more heavily on text features than the N-Gram baseline. With more powerful modeling of text fields, models may begin to rely more heavily on the text fields. An exception here is the *brand_name* feature in the mercari data, but this feature usually contains just a single word in its fields.

---

4. `https://www.machinehack.com/hackathons/product_sentiment_classification_weekend_hackathon_19/overview`. "Anonymous Submission ID 1556" entry.
5. `https://machinehack.com/hackathons/predict_the_data_scientists_salary_in_india_hackathon/overview`. "Xingjian Shi" entry.
6. `https://machinehack.com/hackathons/predict_the_price_of_books/overview`. "Xingjian Shi" entry.
7. `https://www.kaggle.com/c/california-house-prices`, "sxjscience" entry.
8. *Multimodal-Net* achieved a score of 0.38685 on the private leaderboard: `https://www.kaggle.com/c/mercari-price-suggestion-challenge/leaderboard`

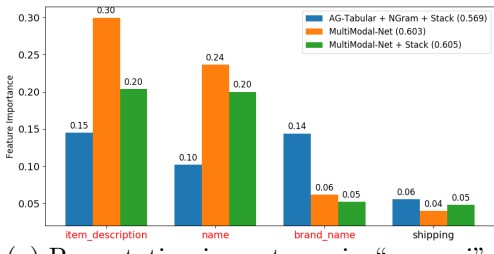

(a) Permutation importance in "mercari".

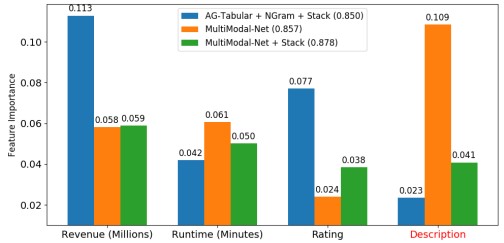

(b) Permutation importance in "imdb".

Figure 3: Importance of text vs. tabular features in two datasets (text features in red).

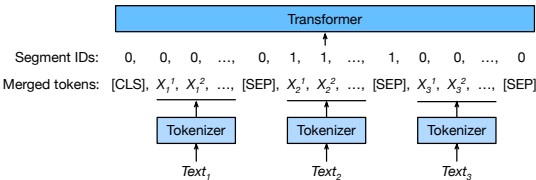

Figure 4: Inputting data from 3 text fields into Transformer.

## Appendix C. Model Details

### C.1 Handling text fields in the Transformer

Given multiple text columns, we feed the tokenized text from all columns jointly into our Transformer, as illustrated in Figure 4. We follow the usual method to format text from multiple passages (Devlin et al., 2019): tokenized inputs from different text fields are merged with special [SEP] delimiter tokens between fields and a [CLS] prefix token is subsequently appended at the start of merged input. To further ensure that the network distinguishes boundaries between adjacent text fields, we alternate 0s and 1s as the segment IDs. Here segment IDs and the [SEP] token were previously used to demarcate boundaries between passages during pre-training (Devlin et al., 2019). After feeding the merged inputs into the Transformer, we can extract its intermediate representations at each position as token-level embeddings (each token has one embedding, which has been contextualized based on information from the other tokens). A single embedding vector for all text fields is obtained from the Transformer's representation at the [CLS] position after feeding the merged input into the network (Devlin et al., 2019). Similarly, just a single text field can be embedded via the Transformer's representation at the [CLS] position, after feeding only this field into the network.

When the total length of tokenized text fields exceed the maximum allowed length (set to be 512 throughout this work), we truncate the input by repeatedly removing one token from the longest individual text field until the length constraint is met. Since self-attention is permutation equivariant, a common practice is to assign an additional vector that encodes each position (namely positional encoding) so that the Transformer can distinguish between identical tokens occurring at different locations (Vaswani et al., 2017). After merging multiple text fields into a single input, we simply assign positional encodings based on this larger input.

## C.2 Network Architectures

In this paper, we used a single-hidden-layer MLP as the basic building block for encoding features and projecting the hidden states. It has one bottleneck layer and uses layer normalization. We use the leaky ReLU activation (with slope set to 0.1) for all basic MLP layers mentioned throughout the paper. For the 6-layer Transformer model in *Fuse-Early*, we used the GeLU activation like Devlin et al. (2019). We set the number of units, heads, and hidden size of FFN (the feedforward layers) in this Transformer to be 64, 4, 256 correspondingly. For the categorical features, we use an encoding network that is similar to the factorized embedding in ALBERT (Lan et al., 2019), in which we use an embedding layer with 32 units and then project it with a basic MLP layer that has 64 bottleneck units. We further set the number of output units in the basic MLP to be the same as the token-embeddings used in the pretrained NLP model (i.e., ELECTRA or RoBERTa) so that all vectors belong to the same space. In the *Fuse-Late* variant, we further concatenate all encoded categorical features and encode them with a second basic MLP layer. Numeric features are concatenated and encoded with one basic MLP layer. These MLP layers all utilize 128 bottleneck units and their output unit number matches the dimensionality of token embeddings for the pretrained Transformer.

## C.3 Neural Network Optimization

All networks are trained with the slanted triangular learning rate scheduler (Howard and Ruder, 2018) with initial learning rate set to 0.0, the maximal learning rate set to $5 \times 10^{-5}$ and warmup set to 0.1. We use a batch size of 128, $10^{-4}$ weight decay, and the AdamW optimizer. All models are trained for 10 epochs and we early stop based on their validation performance. These learning rate and weight decay values were determined via grid search on a single smaller (subsampled) dataset that we used for early initial experiments.

To implement exponential learning rate decay, we set the learning rate multiplier as $\tau^d$ in which $d$ is the layer depth and $\tau$ is the decay factor. The intuition is that the pretrained weights that are closer to the input encode universal properties common across most text and should evolve more slowly during fine-tuning. We set $\tau = 0.8$ in our experiments.

## C.4 Details of AutoGluon Tabular Models in the Stack Ensemble

For improved efficiency, we considered just the following tabular models when running AutoGluon (Erickson et al., 2020):

- Fully-connected Neural Network with ReLU activations (Erickson et al., 2020).

- LightGBM model with default hyperparameters (GBM) (Ke et al., 2017).

- A second LightGBM model with a different set of hyperparameter values. By default, AutoGluon uses this second model in conjunction with the first LightGBM model.

- An implementation of Extremely Randomized Trees from the LightGBM library (Geurts et al., 2006).

- CatBoost gradient boosted trees, which provide sophisticated handling of categorical features (Prokhorenkova et al., 2018)

To avoid overfitting in stacking, all models are trained with 5 fold cross-validation (bagging) as described by Erickson et al. (2020). For classification tasks, the outputs of each base model which are aggregated in the ensemble are taken to be predicted class probabilities.

### C.5 Notes on Hyperparameter Tuning

Note that hyperparameter tuning was not a major focus in this paper. Standard hyperparameter tuning strategies (Shahriari et al., 2015) are readily applicable to our multimodal setting, and the experiments presented here could easily employ the advanced Bayesian optimization techniques available in AutoGluon (Tiao et al., 2020). We expect the performance of all of our proposed AutoML strategies will grow even better with time devoted to hyperparameter tuning. However in this paper we did not conduct such a search and simply used the default hyperparameters supplied by AutoGluon for tabular models, which are already highly performant (Erickson et al., 2020; Fakoor et al., 2020b), and the Transformer hyperparameters are listed here and are viewable in our released code. Over just a few datasets, we found performance did not qualitatively differ with other reasonable hyperparameter settings.

Rather than only reporting a couple thoroughly-tuned results, we instead preferred to spend our time/compute budget to explore more AutoML strategies over more datasets. Note that all H2O AutoML variants reported in Table 2 relied on extensive hyperparameter sweeps (automatically used within H2O), and yet were still unable to outperform our untuned methods. This further supports the claim that we have identified a broadly performant strategy for multimodal AutoML.

## Appendix D. H2O AutoML Baselines

The few other existing tools that aim to automate multimodal text/tabular ML are all commercial software whose source code, allowed scientific usage (benchmarking), and implemented algorithmic strategies remain opaque (Google, 2019; H2O.ai, 2020). As an alternative to AutoGluon, we also run another open-source AutoML tool offered by H2O[9] which is very popular in the data science community. Since H2O AutoML is not designed for the text in our multimodal data tables, we try combining H2O's NLP tool (H2O.ai) and tabular AutoML tool (LeDell and Poirier, 2020).

*H2O AutoML*: We run H2O AutoML directly on the original data of our benchmark. It is assumed that H2O AutoML ignores all text features (as a tabular AutoML framework), but H2O categorizes text vs. other feature types slightly differently than us. For fair assessment, our benchmark leaves key decisions like training/validation splits and how to designate feature types up to each AutoML tool.

*H2O AutoML + Word2Vec*: We featurize text fields via H2O's word2vec algorithm, as described in the H2O.ai tutorial, and then run H2O AutoML on the featurized data.

*H2O AutoML + Pre-Embedding*: We featurize text fields using embeddings from a pretrained ELECTRA Transformer, as in Pre-Embedding, and then run H2O AutoML.

---

9. `https://docs.h2o.ai/h2o/latest-stable/h2o-docs/automl.html`

## Appendix E. Benchmarking Multimodal Text/Tabular AutoML

We aim to design practical systems for real-world data tables that often contain text. The empirical performance of our design decisions is thus what ultimately matters. Representative benchmarks comprised of many diverse datasets are critical for proper evaluation of AutoML, whose aim is to reliably produce reasonable accuracy on arbitrary datasets without manual user-tweaking. While such benchmarks are available for ML with standard tabular data (Gijsbers et al., 2019; Vanschoren et al., 2013; Erickson et al., 2020; Zöller and Huber, 2021), we are not aware of any analogous benchmarks for evaluating multimodal ML. Thus we introduce the first public benchmark[10] for this purpose, which is comprised of 15 tabular datasets, each containing at least one text field in addition to numeric/categorical columns.

Our benchmark strives to represent the types of ML tasks that commonly arise in industry today. In creating the benchmark, we aimed to include a mix of classification vs. regression tasks and datasets from real applications (as opposed to toy academic settings) that contain a rich mix of text, numeric, and categorical columns. Table 3 shows it is comprised of datasets that are quite diverse in terms of: sample-size, problem types, number of features, and type of features. To reflect real-world ML issues, we processed the data minimally (beyond ensuring the labels correspond to meaningful prediction tasks) and thus there are arbitrarily-formatted strings and missing values all throughout. Subsequent accuracy results from Table 2 indicate the 15 underlying prediction problems also vary greatly in terms of both difficulty and how the predictive signal is divided between text/tabular modalities. Given this diversity, systems that can perform well across all 15 datasets are likely to provide real-world value across a broad set of applications.

Each dataset in our benchmark is provided with a prespecified training/test split (usually 20% of the original data reserved for test set). Methods are not allowed to access the test set during training, and for validation (model-selection, hyperparameter-tuning, etc.) instead must themselves hold-out some data from the provided training data. As the choice of training/validation split is a key design decision in AutoML, we leave this flexible for different systems to choose as they see fit. To facilitate comparison between the novel AutoML strategies presented in this paper, we always used the same AutoGluon-provided

---

10. Available at: `https://github.com/sxjscience/automl_multimodal_benchmark`

| Dataset ID | #Train | #Test | #Cat. | #Num. | #Text | Task | Metric | Prediction Target |
|---|---|---|---|---|---|---|---|---|
| prod | 5,091 | 1,273 | 1 | 0 | 1 | multiclass | accuracy | sentiment related to products |
| airbnb | 18,316 | 4,579 | 37 | 24 | 28 | multiclass | accuracy | price of Airbnb listing |
| channel | 20,284 | 5,071 | 1 | 15 | 1 | multiclass | accuracy | category of news article |
| wine | 84,123 | 21,031 | 0 | 2 | 3 | multiclass | accuracy | variety of wine |
| imdb | 800 | 200 | 0 | 7 | 4 | binary | roc-auc | whether film is a drama |
| jigsaw | 100,000 | 25,000 | 2 | 27 | 1 | binary | roc-auc | whether comments are toxic |
| fake | 12,725 | 3,182 | 2 | 0 | 3 | binary | roc-auc | whether job postings are fake |
| kick | 86,502 | 21,626 | 3 | 3 | 3 | binary | roc-auc | will Kickstarter get funding |
| ae | 22,662 | 5,666 | 3 | 2 | 6 | regression | $R^2$ | American-Eagle item prices |
| qaa | 4,863 | 1,216 | 1 | 0 | 3 | regression | $R^2$ | type of answer |
| qaq | 4,863 | 1,216 | 1 | 0 | 3 | regression | $R^2$ | type of question |
| cloth | 18,788 | 4,698 | 2 | 1 | 3 | regression | $R^2$ | review score |
| mercari | 100,000 | 25,000 | 3 | 0 | 6 | regression | $R^2$ | price of Mercari products |
| jc | 10,860 | 2,715 | 0 | 2 | 3 | regression | $R^2$ | price of JC Penney products |
| pop | 24,007 | 6,002 | 1 | 2 | 1 | regression | $R^2$ | news article popularity online |

Table 3: The 15 multimodal datasets that comprise our benchmark. '#Cat.', '#Num.' and '#Text' count the number of categorical, numeric, and text features in each dataset. In PDF, you can click on each Dataset ID for link to original data source.

training/validation split, which is stratified based on labels in classification tasks. Our use of other AutoML frameworks beyond AutoGluon (e.g. H2O) allows each framework to choose their own data splitting scheme.

## E.1 Descriptions of each Dataset

**prod**: Classify the sentiment of user reviews related to products based on the review text and product type.

**airbnb**: Predict the price label of AirBnb listings in Melbourne, Australia based on the page of each listing which includes many miscellaneous features about the listing.

**channel**: Predict which news category (i.e. channel) a Mashable.com news article belongs to based on the text of its title, as well as auxiliary numerical features like the number of words in the article, its average token length, how many keywords are listed, etc.

**wine**: Classify the variety of wines based on tasting descriptions from sommeliers, their price, country-of-origin, and other features.

**imdb**: Predict whether or not a movie falls within the Drama category based on its name, description, actors/directors, year released, runtime, and other features.*

**jigsaw**: Predict whether online social media comments are toxic based on their text and additional features related to the poster.

**fake**: Predict whether online job postings are real or fake based on their text, amount of salary offered, degree of education demanded, etc.

**kick**: Predict whether a proposed Kickstarter project will get funding based on its title, description, amount of money requested, date posted, and other features.

**ae**: Predict the price of inner-wear items sold by retailer American Eagle based on features from their online product page.*

**qaa**: Given a question and an answer (from the Crowdsource team at Google) as well as additional category features, predict the type of the answer in relation to the question.

**qaq**: Given a question and an answer (from the Crowdsource team at Google) as well as additional category features, predict the type of question in relation to the answer.

**cloth**: Predict the score of a customer review of clothing items (sold by an anonymous retailer) based on the review text, and product features like the clothing category.

**mercari**: Predict the price of items sold in the online marketplace of Mercari based on miscellaneous information from the product page like name, description, free shipping, etc.

**jc**: Predict the sale price of items sold on the website of the retailer JC Penney based on miscellaneous information on the product page like its title, description, rating, etc.*

**pop**: Predict the popularity (number of shares on social media, on log-scale) of Mashable.com news articles based on the text of their title, as well as auxiliary numerical features like the number of words in the article, its average token length, and how many keywords are listed, etc.

---

∗. PromptCloud released the original version of the data from which the version of this dataset in our benchmark was created.

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
