# OpenReview forum: "Multimodal AutoML on Structured Tables with Text Fields"
_ICML.cc/2021/Workshop/AutoML — AutoML@ICML2021 Oral_

### Official Review · Reviewer_mMWg · 2021-06-12
**Good Paper with Open Source Code**

**Rating:** 8
**Confidence:** 2

**Review:**

The paper presents an AutoML system for tabular data with text fields. Furthermore, the paper introduces a benchmark for tabular data with text fields and evaluates the AutoML system successfully on this benchmark but also on two competitions and a Kaggle Challange. The benchmark and the code for the system are available.

The paper uses pre-trained Transformer models to build a multimodal model. The paper describes and evaluates multiple ways to embed text and combine text and tabular data. The methods are well-written and comprehensible.

The experiment is based on the introduced benchmark and contains a detailed comparison of AutoML strategies on tabular data with text fields.

The paper lacks a conclusion and could analyze more the difference between pre-embedding and n-grams/word2vec.

---

### Official Review · Reviewer_vq89 · 2021-06-16
**Interesting extension of AutoML to text features**

**Rating:** 8
**Confidence:** 3

**Review:**

This paper proposes the use of modern NLP architecture to handle text fields in tabular data for AutoML.

Positives:

- The authors provide a new benchmark set of 15 datasets with a focus on text fields
- Solid ablation analysis of separate components of their approach
- Clearly and well written

Negatives (all rather minor):

- I am a bit unsure how much I dislike averaging over different performance measures (R2, AUC, ACC) and all datasets without any form of scaling in the benchmark.
- Only one other AutoML tool is used for comparison, maybe add a sentence why you selected H2O as a baseline.



Minor comments:
- In Sec 2. Methods: The figure reference for fuse-late in the text should be 1c) instead of 1d)


Overall I really enjoyed reading the paper. The addressed problem is of high practical relevance and their approach is very reasonable. The publishing of their benchmarks allows for easy and fair comparison of potential further work.

---

### Decision · Program_Chairs · 2021-06-21

Accept (Oral)